# Coinage Metal-Catalyzed Asymmetric Reactions of *ortho*-Alkynylaryl and Heteroaryl Aldehydes and Ketones

**DOI:** 10.3390/molecules27206970

**Published:** 2022-10-17

**Authors:** Romain Melot, Véronique Michelet

**Affiliations:** Université Côte d’Azur, Institut de Chimie de Nice, Valrose Park, 06108 Nice, France

**Keywords:** asymmetric catalysis, coinage metals, copper, gold, isochromenes, silver

## Abstract

Coinage metals have become the metal of choice due to their excellent catalytic activity in organic transformation processes. Combining various chiral ligands and coinage metals became a productive area of research and access to heterocyclic derivatives according to an efficient and sustainable manner. This review was devoted to the various recently developed coinage metal-catalyzed domino processes of *ortho*-alkynylaryl and heteroaryl aldehydes and ketones leading to functionalized heterocycles. Various gold chiral complexes were presented, and methods of preparations of chromenes along with indoles were covered. Ag-chiral complexes are also prone to interesting activities such as cyclization followed by reduction and functionalization with enolizable ketones or (diazomethyl)phosphonate. Asymmetric Cu-catalyzed domino cyclization and asymmetric transfer hydrogenation reactions efficiently led to functionalized chromenes. Some remarkable examples involving copper associated with ruthenium in the context of a cyclization and asymmetric hydrogenation process were also presented.

## 1. Introduction

Transition metal-catalyzed reactions of carbonyl derivatives bearing an ortho-substituted carbon–carbon triple bond have been widely studied and allowed highly interested domino processes in the literature [1,2,3,4,5,6,7,8,9,10,11,12,13,14,15]. The first inspiring example of such process was reported by Yamamoto’s group in 2002 and proposed the palladium (II)-catalyzed acetalization–cycloisomerization reactions on aldehyde-alkynes, in the presence of *p*-benzoquinone and alcohol (Figure 1) [16,17,18].

Other metals such as Pd [19,20,21,22], Ru [23], Au [24,25,26,27], Ag [28,29,30], Cu [31,32], and electrophiles have then been engaged, and two pathways have been identified depending on the nature of the catalyst, 5-*exo-dig* and 6-*endo-dig*, resulting in various cyclic products, with the outcome of the process being mainly dependent on the carbophilic or oxophilic properties of the catalytic species (Figure 2).

Following the electrophilic cyclization of *o*-alkynylaryl carbonyl compounds or the oxo-activation of carbonyl moieties, nucleophile addition reactions have been demonstrated and led to the 1-substituted 1*H*-isochromene derivatives [33]. These domino transformations may involve alcohol, acid, aromatic ring, alkyne, alkene, and diethyl phosphite derivatives as partners, but few of them have been successfully proposed according to an asymmetric way [34,35,36]. This review proposed an overview of enantioelective domino transformations of *ortho*-alkynylaryl and heteroaryl aldehydes and ketones in the presence of transition metal. We will first focus on gold catalysis as the historical first developed alkoxylation–cyclization process, and then we will present the silver- and copper-catalyzed reactions.

## 2. Gold Catalysis

Gold catalysts are known to effectively promote intramolecular *ortho*-alkynylbenzaldehyde cyclization with concomitant addition of different nucleophiles [37,38,39,40,41,42]. However, the development of an asymmetric version of such tandem process is relatively recent. Indeed, in 2012, Slaugther and coworkers disclosed the first enantioselective *ortho*-alkynylbenzaldehyde cyclization using a chiral gold(I) complex as the catalyst [43]. This work began by the design and synthesis of a new class of chiral gold(I) acyclic diaminocarbene (ADC) complexes based on binaphtyl scaffold (Figure 3). The authors began the synthesis by the preparation of three different 2-isocyanobinaphtyl from BINOL based on literature precedents. Metalation of these compounds with Au(THT)Cl (THT: tetrahydrothiophene) provided the corresponding Au(I) complexes. Finally, the reaction of the obtained gold(I) complexes with a bulky secondary amine such as diisopropylamine led to the formation of the desired chiral gold(I) ADC complexes. X-ray crystal structures of the prepared complexes revealed a *syn* relationship between gold and the binaphtyl unit. Moreover, the presence of a 3,5-(CF_3_)_2_C_6_H_5_ moiety in the 2′-position of the binaphtyl part allowed an additional weak gold-π interaction, which was not present in the case of a phenyl substituent in the same position. This interaction subsequently creates more pronounced chiral surroundings at the metal center. With these new chiral catalysts on hand, the authors chose to investigate the development of an asymmetric *ortho*-alkynylbenzaldehyde cyclization with the simultaneous addition of alcohol. After a fine optimization, the best conditions for this reaction were found.

Complex A3, exhibiting a gold–*π* interaction, combined with LiNTf_2_ as a halogen scavenger in DCE (1,2-dichloroethane) enabled cyclization with nucleophilic attack of different alcohols under mild conditions with a high level of enantioselectivity (Figure 4a). However, limitations were observed with alkyne substrates bearing an alkyl substituent or when shorter primary alcohols were used as nucleophiles. To address this issue, the author prepared a new catalyst (B3) using the chiral amine ((*S*)-PhMeCH)_2_NH in place of *i*-Pr_2_NH. This modified catalyst allowed the extension of the scope to less reactive substrates and alcohols (Figure 4b). Moreover, catalyst B3 led to a higher level of enantioselectivity on one example already prepared with catalyst A.

In complement, DFT (density functional theory) calculations were performed on the rotamers of all catalysts. The obtained results indicate a significantly stabilized rotamer (8 kJ.mol^−1^) when a 3,5-(CF_3_)_2_C_6_H_5_ moiety was in proximity to gold supporting the X-ray crystallographic data. These combined results confirm a cation–π interaction between gold and the electron-deficient aryl group.

More recently, Wong and coworkers described the synthesis of a new class of BINOL–gold(III) complexes (BINOL: 1,1′-Binaphthalene-2,2′-diol) [44]. The latter were obtained by the reaction of BINOL with an oxazoline-based gold(III) dichloride complex, which were prepared based on literature precedents. Unexpectedly, instead of an envisaged O,O’-chelation mode, the authors observed the clean formation of a complex where BINOL adopted an unusual C,O-chelation (Figure 5). This was confirmed by a new carbonyl signal in ^13^C NMR and X-ray crystallographic analysis. Remarkably, this transformation occurs with complete axial-to-central chirality transfer.

This reactivity was then extended to a large variety of BINOL derivatives and cyclometalated (C^N)gold(III) dichloride complexes (Figure 6). The variation of the cyclometalated part was first studied.

Chiral oxazolines were well-tolerated without any influence of the existing stereocenter on the chirality transfer, allowing the access to all possible stereoisomers. Moreover, pyridine-based complexes with different linkers can also be prepared in excellent yields using this strategy. Then, the authors evaluated the reaction outcome with a variety of substituted BINOLs. The presence of methyl, triphenylsilyl, aromatic, or bromide groups in 3,3′ or 6,6′ positions results in clean formation of the expected complexes. Once again, the chirality on the oxazoline part did not influence the chirality transfer with both enantiomers of 3,3′-Me_2_BINOL as a reaction partner. Of note, the authors also described the reaction of (rac)-BINOL with a chiral oxazoline-based gold(III) dichloride complex. In this case, two diastereoisomers were generated and separated by silica gel column chromatography. This strategy constitutes an elegant way to access two new chiral gold(III) complexes from inexpensive racemic BINOL.

The authors then evaluated the catalytic activity of the newly prepared Au(III) complexes for an intramolecular cyclization of *ortho*-alkynylbenzaldehydes. These organometallic species allowed the effective formation of 1*H*-isochromene derivatives in good yields from trialkyl orthoformates and o-alkynylbenzaldehydes (Figure 7, left). In this case, the trialkyl orthoformate probably generates the corresponding free alcohol participating in this tandem process. Interestingly, when *D*-(*+*)-camphorsulfonic acid (CSA) was added to the reaction mixture, formation of 3-alkoxyindanones was observed (Figure 7, right).

This drastic change in the product distribution can be explained by an in situ acetal formation with concomitant carboalkoxylation based on the work of Toste and coworkers (Figure 8) [45].

Preliminary studies of Wong’s group in the same communication showed that a chiral BINOL–gold(III) complex (B9) as a catalyst allowed moderate enantioselectivity for the reaction of *ortho*-alkynylbenzaldehydes and methanol (Figure 9).

Of note, in 2019, the same research group reported the carboalkoxylation of *ortho*-alkynylbenzaldehyde dimethyl acetals using chiral cyclometalated oxazoline gold(III) catalysts [46]. Although interesting enantioselectivities were observed, the use of an unmasked aldehyde was not reported in this work.

More recently, our group became interested in the development of a domino cycloisomerization and alkoxylation of propargylated indole carbaldehydes with the addition of nucleophilic alcohols [47]. Indeed, such transformation allowed an access to valuable oxazinoindole derivatives from readily accessible precursors. After careful optimization, IPrAu(MeCN)BF_4_ (IPr: 1,3-bis(2,6-diisopropylphenyl)imidazole-2-ylidene) as a catalyst gave the best results using only two equivalents of alcohols as a partner in DCE at -20 °C. This set of conditions permits a smooth reaction of different propargylated indole carbaldehydes with a variety of alcohols (Figure 10). Both alkyl and aryl groups were well-tolerated as a substituent of the alkyne part. When methyl or halide groups were introduced in position 4 or 6 of the indole, the reaction still smoothly proceeded. Moreover, several primary alcohols can be used as a nucleophilic partner such as methanol, allyl alcohol, (*S*)-3-methylbutanol, or geraniol. Then, a range of chiral gold(I) catalysts were tested to develop an asymmetric version of this transformation. These experiments indicated that catalysts that constitute of ligands from the MeO-BIPHEP (2,2′-bis(diphenylphosphino)-6,6′-dimethoxy-1,1′-biphenyl) and SEGPHOS (5,5′-bis(diphenylphosphino)-4,4′-bi-1,3-benzodioxole) families induce good enantioselectivities. Further investigations revealed that catalytic amounts of DTBM-SEGPHOSAu_2_Cl_2_ (DTBM-SEGPHOS: 5,5′-bis[di(3,5-di-tert-butyl-4-methoxyphenyl)phosphino]-4,4′-bi-1,3-benzodioxole) and silver hexafluoroantimonate in DCE at -20 °C effectively promote the reaction, leading to an improved enantiomeric excess. Gratifyingly, the addition of 4Å molecular sieves results in the best enantioselectivity, even though this improvement remains unclear so far. A range of substrates was then engaged under these optimized conditions (Figure 11). An aryl group as an alkyne substituent reliably led to very good enantiomeric excess ranging between 79% and 86% with different nucleophilic alcohols. Moreover, substituents at the indole part did not significatively influence the asymmetric induction. Nevertheless, the enantioselectivity decreased with an ethyl group on the alkyne or when methanol was used as a nucleophilic partner.

## 3. Silver Catalysis

Traditionally, silver complexes have been extensively exploited in organometallic transformations implying *ortho*-alkynylaryl and heteroaryl aldehydes and ketones, as pioneered by Belmont’s and Abbiati’s groups [8,27,28,29,39]. The development of enantioselective transformations involving silver complexes has been described by Terada’s group in 2014 [48]. It was particularly challenging to control chirality on the reactive intermediate, e.g., the isobenzopyrylium ion. The authors anticipated that the formation of ion-pair salts between the isobenzopyrylium ion and a less nucleophilic counteranion could induce an enantioselective process. The reaction conditions, implying the mode of cyclization, were optimized, and silver was the most selective metal catalyst employed (with copper). The desired 6-*endo* derivative was isolated in 96% yield in the presence of AgOTf and Cu(OTf)_2_, with a higher activity with silver salts (Figure 12).

The enantioselective variant of the reaction was then optimized in the presence of a chiral silver phosphate, obtained from the combination of phosphoric acid and Ag_2_CO_3_. Classical binol-derived phosphoric acids were evaluated, and the best one was that conducted with the pentafluorophenyl-substituted one at the 3- and 3′-positions of the binaphthol backbone, leading to 85% enantiomeric excess and 87% when 5 Å MS was used as an additive. The chemical yield was still high in both cases (95–97% isolated yield). The scope and limitations were then studied and implied a broad range of alkynylaryl ketones; some examples are proposed in Figure 13. Generally, the reactivity and enantioselectivity were highly dependent on the substrate and nature of the substituents. In the case of alkyl-substituted ketones and aryl-functionalized alkynes, the isochromene derivatives were obtained in high yields and good enantioselectivities. It is noteworthy that the presence of an electron-withdrawing group in the *para* position on the aromatic group at the alkyne position led to a slight decrease in enantioselectivity as compared with that observed with the phenyl substrate. The introduction of an alkyl substituent at the alkynyl position afforded a large decrease in the enantioselectivity, as the *^n^*Bu isochromene was isolated in 22% *ee*. Changing the methylketone to another linear or branched alkyl ketone also underwent the reaction with a slight reduction in enantioselectivity. The presence of a fluorine atom on the aromatic ring bearing the ketone did not change the result of the process and led to 87% and 90% ee, respectively. The best results were obtained in the case of substrates bearing aromatic substituents both at the keto position as well as the alkynyl position (Ph), and the ee values of the isochromenes were all around 90% without molecular sieves in these cases.

The group of Enders also became interested in such process in the presence of enolizable ketones such as other nucleophiles than the Hantzsch ester (HEH) [49]. The 2-alkynylbenzaldehyde was reacted with acetylacetone in the presence of silver carbonate as a catalyst and 10 mol% of Brönsted acid. The optimization of the reaction conditions (Brönsted acid, solvent, and temperature) showed that the use of diphenyl phosphate (DPP) in *n*-hexane at 40 °C led to the desired functionalized isochromene in 93% yield (Figure 14, Reaction 1). The authors then tested a wide range of substituted *o*-alkynylarylaldehydes in the presence of several ketones such as phenyl acetone, 2-phenyl cyclohexanone, and methyl cyclopentanone-2-carboxylate. The reaction outcome was, as expected, influenced by the enolization ability of the ketone, and a longer reaction time was needed in the case of less-enolizable ketones. A single example of enantioselective reaction was reported based on Terada’s previous report (Figure 14, Reaction 2). The use of TRIP (3,3′-Bis(2,4,6-triisopropylphenyl)-1,1′-bi-2-naphthol cyclic monophosphate) as chiral phosphoric acid instead of the achiral diphenyl phosphate led to the enantiomerically enriched 1*H*-isochromene, in the presence of 2.5 mol% of Ag_2_CO_3_ as the silver catalyst and 2-phenyl cyclohexan-1-one as the nucleophile. The reaction was conducted at 40 °C for 30 min and then stirred at 0 °C for 3 days leading to the desired compound in 33% yield, a dr of 2.3:1 and 63% *ee* of the major diastereomer.

More recently, the group of Peng developed a domino process implying *o*-alkynylacetophenones with (diazomethyl)phosphonate for the synthesis of functional isochromenes [50]. Interestingly, (diazomethyl)phosphonates known as carbene precursors could efficiently react with an in situ generated isobenzopyrylium ion of *o*-alkynylacetophenone (Figure 15). The asymmetric variant was optimized in the presence of various chiral bis(oxazoline) ligands, silver salts, and solvents. The best conditions described in Figure 14 involved AgOTf as the silver salt and the adamantyl-based chiral bis-oxazoline, 4 Å MS, as an additive in chlorobenzene at 0 °C. The resulting isochromene was isolated in 98% yield and 94% enantiomeric excess.

The authors then proposed a large scope of their reaction conditions by changing the groups on the alkyne and aromatic ring (Figure 16). Generally, the keto derivatives bearing electron-donating substituents on the aryl moiety at the alkyne terminus led to the desired isochromenes in good-to-excellent yields and enantioselectivities (>90% *ee*). Electron-withdrawing groups induced a decrease in yields but generally allowed good enantioselectivities. The authors explained the lower reactivity by a lower stability of the in situ formed isobenzopyrylium ion. The substitution pattern has a great influence on the reactivity. The influence of various groups on the aromatic moiety bearing the ketone was also investigated. Interestingly, the keton-ynes with electron-donating and electron-withdrawing groups were transformed to functionalized isochromenes in excellent yields and good-to-excellent enantioselectivities (85%−94% *ee*). Few limitations have been observed: in the case of the substrate with the chlorine atom at the 5-position, only 21% of the product was obtained in 84% *ee*. This result may be explained by a lower electron density on the carbonyl compound compared with similar substrates with such groups located at the *para* position. When the methyl ketone was replaced by an ethyl derivative, a modest yield (47%) and good ee (93%) were observed. The reaction mechanism was investigated by DFT calculations.

The results of the X-ray crystallographic analysis of a methoxy-substituted derivative confirmed the absolute configuration of the bicyclic compound and, therefore, the sense of the stereoselectivity. Considering the interest of phosphine-containing heterocycles, the authors showed that the enol ether could be reduced as well as transformed to a hydrazone or diazo compound. The bicyclic hydrazone was also converted to a tricyclic derivative via a Bi(OTf)_3_-catalyzed hydroamination.

## 4. Copper Catalysis

Apart from gold and silver, copper was also an efficient metal for domino reaction of carbonyl-yne derivatives. In the same concept as Terada’s work developed a few months later, Akiyana developed an asymmetric synthesis of enantiomerically enriched isochromenes through the copper(II)/phosphate-catalyzed intramolecular cyclization–asymmetric transfer hydrogenation reaction of *o*-alkynylacetophenone derivatives [51]. Several cationic transition-metal–phosphate catalysts, prepared from a transition-metal salt, 3,4,5-trifluorophenyl-substituted chiral binaphthyl phosphoric acid, and silver carbonate, were engaged with the carbonyl-yne derivative in the presence of 4 Å molecular sieves and the Hantzsch ester as a hydride donor (Figure 17).

This sequence allows the synthesis of aryl-substituted isochromenes containing various substituents in 75–90% yields and good-to-excellent enantioselectivities up to 97% (Figure 18). The reaction conditions are compatible with chlorine, bromine, methoxy, and methyl groups on the aromatic ring of the alkyne. Me-ketones were generally employed, but an example with an ethyl group afforded the desired bicyclic compounds in 84% yield and a 90.1:9.9 enantiomeric ratio.

Instead of using Hantzsch ester, the group of Yu and Fan found that the cationic ruthenium complexes of chiral monosulfonated diamines were very efficient catalysts for the asymmetric hydrogenation of isochromenyliums [52]. Very elegantly, they found compatible catalysts for the overall chemoselective and enantioselective process, starting from keto-ynes derivatives. A thorough optimization led to the association of copper and ruthenium in low catalytic loadings, with a clear influence of the ratio of the Cu and Ru catalysts, the *N*-sulfonate substituent, and the solvent (GDME: ethylene glycol dimethyl ether). The Me-substituted adduct was isolated in >95% enantiomeric excess and 91% yield (Figure 19).

Under the optimized reaction conditions, a variety of *ortho*-(alkynyl)aryl ketones were engaged in the tandem reaction giving the desired *R*-configured 1*H*-isochromenes in excellent yields and enantioselectivities (Figure 20). The aryl group at the alkyne terminus could be substituted by electron-donating or electron-withdrawing groups at the *para* position, still affording high yields and excellent enantioselectivities. The high yields and enantioselectivities were also maintained when linear or branched alkyl chains on the ketone moiety were present. The introduction of a fluorine atom on the tethered aryl ring induced a slight decrease in the enantiomeric excess. The case of alkyl-substituted alkynes was particularly interesting as they are rare in the literature. Propyl and butyl chains were tolerated, whereas *tert*-butyl and a longer chain such as *n*-octyl afforded the desired products with slightly lower but still good enantioselectivities.

The interest of this catalytic asymmetric tandem reaction was further demonstrated by the scale-up synthesis of two 1*H*-isochromenes and by performing the hydrogenation of the remaining double bond under diastereoselective classical heterogeneous conditions. The authors also became interested in the mechanism of this tandem process, showing via deuterium labeling as well as asymmetric hydrogenation of a stable isochromenylium salt in the presence of a Ru hydride complex. DFT calculations were performed, allowing further insight into the origin of the enantioselectivity.

## 5. Conclusions

Few examples have been described so far for the transformations of keto-yne derivatives to functionalized isochromenes. The main challenging step is the reduction and/or functionalization of the isochromenylium intermediate that can be generated in the presence of gold, silver, and copper salts. The choice of the chiral ligands is generally dictated by the metal, from atropisomeric phosphanes, carbenes, to phosphoric acids. A remarkably achiral copper complex could be associated with cationic ruthenium complexes of chiral monosulfonated diamines for the hydrogenation step. Considering the large number of nucleophiles suitable for domino processes involving *ortho*-alkynylaryl and heteroaryl aldehydes as well as the interest of isochromenes [53,54,55,56,57,58,59], one may expect and hope for other examples to be described.

## Data Availability

Not applicable.

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
