# Peer review of "Coinage Metal-Catalyzed Asymmetric Reactions of ortho-Alkynylaryl and Heteroaryl Aldehydes and Ketones"

_molecules, 2022, doi:10.3390/molecules27206970_

Round 1

Reviewer 1 Report

This review showed asymmetric reaction of o-alkynylarylketones/aldehydes catalyzed by coinage metal to give isochemoenes which sound interesting.

Unfortunately, In 2020, Chauhan and co-workers published the review on “Catalytic Asymmetric Synthesis of Isochroman Derivatives” which included asymmetric synthesis of isochemoenes (10.1002/adsc.202000243) and almost the reaction that the author present were reviewed already in that review except the author work in JOC 2021. So, the author should try other ways to present this reaction.

This manuscript is not well organized and introduction part does not convince the reader.

Author Response

This reference was added, we are sorry for not having seen it before. Our presentation is different than the review from Chauhan, which makes it complementary too.

Reviewer 2 Report

The authors have been submitted a review paper in asymmetric reactions of alkynylaryl and heteroaryl aldehydes and ketones using Ag, Au or Cu catalysts. I can recommend publication of this article in the Molecules journal conditional on addressing the minor points below. The subject of the paper is fall within the scope of the journal, especially the domino cyclization reactions, asymmetric synthesis of heterocycles, which should be useful to scientists attempting to prepare these scaffolds.

The following items should be addressed:

Please try to explain why the mentioned metal atom is the best for the presented reaction. What is the reason of their effectivity compared to others?

Please explain the role of the metal atoms, especially when at least 2 different ones were used.

A few additional remarks, mistakes:

In row 32, please cite the cyclisation types as Baldwin, J. E. Rules for Ring Closure. J. Chem. Soc. Chem. Commun. 1976, 18, 734736,  DOI: 10.1039/c39760000734

A minor mistake in row 32: 1H-isocromenes or 1H-isocromene derivatives

In row 56, please define THT.

In row 70, “Complex A” instead of “Complexes A”

In row 91. This was confirmed

In scheme 6, right, bottom, please replace both Br to R.

In row 114, an elegant way

For easier follow-up, please draw the plausible mechanism of the formation of indanones in Scheme 7.

Please mark the stereochemistry of the product in Scheme 8.

In row 149, “allyl alcohol”

Please define DTBM, BIPHEP, SEGPHOS, HEH, etc. abbreviations in the text

In row 182. It is not clear what does “Table 2. CO3” mean. Maybe some residue.

In row 194. “alynyl position”? alkynyl?

In row 247. Something is missing. Lower electron density?

In row 303. Some extra ? was added, please remove them.

General.

Please insert a space between number and unit of measure.

Please double check the plural and singular conjugation in the whole text.

There are several Complex/catalyst A and B in the manuscript. Please create a clear nomenclature like Complex A1 in scheme 1, Complex A2 in scheme 2, etc.

Please use ee values in all the schemes (Scheme 16, 17, etc.) instead of er.

Author Response

Reviewer #1:

The authors have been submitted a review paper in asymmetric reactions of alkynylaryl and heteroaryl aldehydes and ketones using Ag, Au or Cu catalysts. I can recommend publication of this article in the Molecules journal conditional on addressing the minor points below.…

Please try to explain why the mentioned metal atom is the best for the presented reaction. What is the reason of their effectivity compared to others?

It is difficult to add some information about from oxophilic and carbophilic properties. We have not changed as we have added the key reference from Baldwin and have modified (clarified) scheme 2 and added a scheme in introduction.

Please explain the role of the metal atoms, especially when at least 2 different ones were used. We have detailed the role of Ru and Ag when they are used. We have added a scheme with mechanism (Scheme 8).

A few additional remarks, mistakes:

In row 32, please cite the cyclisation types as Baldwin, J. E. Rules for Ring Closure. J. Chem. Soc. Chem. Commun. 1976, 18, 734– 736,  DOI: 10.1039/c39760000734

Reference 33 was added.

A minor mistake in row 32: 1H-isocromenes or 1H-isocromene derivatives Corrected 

In row 56, please define THT. This was defined.

In row 70, “Complex A” instead of “Complexes A” This has been changed to complex A3 in accordance with general comments of the same reviewer

In row 91. This was confirmed This typo was corrected.

In scheme 6, right, bottom, please replace both Br to R. This typo was corrected.

In row 114, an elegant way This typo was corrected.

For easier follow-up, please draw the plausible mechanism of the formation of indanones in Scheme 7. A scheme with a plausible mechanism has been added.

Please mark the stereochemistry of the product in Scheme 8. The authors (Wong and coworkers) did not provide this information.

In row 149, “allyl alcohol” This typo was corrected.

Please define DTBM, BIPHEP, SEGPHOS, HEH, etc. The abbreviations in the text have been added.

In row 182. It is not clear what does “Table 2. CO3” mean. Maybe some residue. This was not in our submitted version, a full sentence was cut and change for “Table 2. CO3” ???

In row 194. “alynyl position”? alkynyl? We changed for alkynyl.

In row 247. Something is missing. Lower electron density? This typo was corrected.

In row 303. Some extra ? was added, please remove them. This typo was corrected.

General.

Please insert a space between number and unit of measure. This was corrected.

Please double check the plural and singular conjugation in the whole text. We have checked all sentences.

There are several Complex/catalyst A and B in the manuscript. Please create a clear nomenclature like Complex A1 in scheme 1, Complex A2 in scheme 2, etc. This was done.

Please use ee values in all the schemes (Scheme 16, 17, etc.) instead of er. The authors have used er, but we have changed to ee as suggested.

Reviewer 3 Report

Michelet and co-worker reported a nice overview on recent asymmetric transformations of alkynyl aldehydes using gold, silver and copper catalysis. The work is well written and deals with important synthetic transformations that certain deserve publication in Molecules. However, some minor concerns should be addressed to improve the overall high quality of this manuscript:

1) Introduction: line27. A scheme on the first example reported by yamamoto might be helpful for the reader.

2) Missing leading reviews on asymmetric transformations with gold, copper and silver. Please implement. As in example: Chem. Rev. 2016, 45, 4567; Chem. rev. 2016, 116, 14868; Tetrahedron 2021, 93, 132238. 

3) Scheme 1 is somehow not clear. Please, consider to re-draw in a more concise manner.

4) Scheme 5, 6. Of course the synthesis of these chiral gold catalysts is of highly importance but their description is out of the scope of this review. I suggest to use only one scheme adding only few selected examples.

5) Scheme 9. Add the structure of IPr ligand. 

6) Scheme 10. L(AuCl)2 is more appropriate than L(Au2Cl2).

Author Response

Michelet and co-worker reported a nice overview on recent asymmetric transformations of alkynyl aldehydes using gold, silver and copper catalysis. The work is well written and deals with important synthetic transformations that certain deserve publication in Molecules. However, some minor concerns should be addressed to improve the overall high quality of this manuscript:

1) Introduction: line27. A scheme on the first example reported by yamamoto might be helpful for the reader. A scheme has been added.

2) Missing leading reviews on asymmetric transformations with gold, copper and silver. Please implement. As in example:

  • Zi, W.; Dean Toste, F. Recent Advances in Enantioselective Gold Catalysis. Chem. Soc. Rev. 2016, 45 (16), 4567-4589. https://doi.org/10.1039/C5CS00929D;
  • Pellissier, H. Enantioselective Silver-Catalyzed Transformations. Chem. Rev. 2016, 116 (23), 14868-14917. https://doi.org/10.1021/acs.chemrev.6b00639;
  • Shah, S.; Das, B. G.; Singh, V. K. Recent Advancement in Copper-Catalyzed Asymmetric Reactions of Alkynes. Tetrahedron 2021, 93, 132238. https://doi.org/10.1016/j.tet.2021.132238.

References 34-36 were added.

3) Scheme 1 is somehow not clear. Please, consider to re-draw in a more concise manner. A revised version of the scheme has been provided.

4) Scheme 5, 6. Of course the synthesis of these chiral gold catalysts is of highly importance but their description is out of the scope of this review. I suggest to use only one scheme adding only few selected examples. Scheme 5 & 6 were merged to build a global scheme.

5) Scheme 9. Add the structure of IPr ligand. The structure was added.

6) Scheme 10. L(AuCl)2 is more appropriate than L(Au2Cl2). This was corrected.

Round 2

Reviewer 1 Report

This manuscript has been improved and the quality is much better than previous version.